# Impaired Quality of Life in Croatian IBD Patients in the Era of Advanced Treatment Options

**DOI:** 10.3390/healthcare13141681

**Published:** 2025-07-12

**Authors:** Alen Bišćanin, Leon Palac, Zdravko Dorosulić, Dominik Kralj, Petra Ćaćić, Filip Babić, Doris Ogresta, Davor Hrabar, Vedran Tomašić

**Affiliations:** 1Department of Gastroenterology, Sestre Milosrdnice University Hospital Centre, 10000 Zagreb, Croatia; alen.biscanin@gmail.com (A.B.); zdorosulic@yahoo.com (Z.D.); dominik.rex@gmail.com (D.K.); petra.cacic@gmail.com (P.Ć.); filip.babic18@gmail.com (F.B.); doris.ogresta@gmail.com (D.O.); davor.hrabar@kbcsm.hr (D.H.); tomasicvedran@gmail.com (V.T.); 2School of Medicine, University of Zagreb, 10000 Zagreb, Croatia

**Keywords:** IBD, quality of life

## Abstract

**Background/Objectives:** Inflammatory bowel disease (IBD) is a chronic gastrointestinal disorder marked by relapsing episodes of gastrointestinal inflammation, potentially causing severe symptoms. These unpredictable acute episodes, paired with chronic disabilities, such as fatigue and malabsorption, and extensive pharmacological and surgical treatments, can severely impact patients’ quality of life. This study aimed to assess which aspects of the patients’ lives IBD impacts, and how IBD patients perceive their disease. **Methods:** All IBD patients who had an appointment in our tertiary centre from 10 October 2022 to 21 February 2023, were invited to complete anonymous questionnaires. The questionnaires used were IBDQ-32, WPAI, and IBD Disk, all designed specifically to assess the IBD patients’ quality of life. **Results**: The questionnaires were completed by a total of 159 participants, 51% of whom were males, 47.9% who had UC, and 49.4% who had been or were currently treated with biologics. There was no statistically significant difference in the answers from patients with CD compared to UC, as well as those treated with conventional therapies compared to those with advanced options. Most of them considered their health to be good, but only a few (12.8%) claimed, with absolute certainty, that their health was at the level of healthy individuals, and only 13 (8.3%) claimed their health was excellent. A total of 95 (60.1%) participants expressed at least minor limitations when performing strenuous activities, but lighter forms of activities were not affected as much by the disease. A significant portion (48.7%) of the participants believed they were exposed to more stress than others, and their current pharmacological therapy was the cause of fear in 26.5%. A total of 119 (75.3%) participants believed that the disease affected their lives at least mildly during remission. **Conclusions**: Our study showed that IBD patients have diminished quality of life, not only in the periods of active disease but also during clinical remission. The decline in quality of life was not solely attributed to physical symptoms, as previously thought. Other factors, such as mental health issues, were found to impact quality of life as well. We firmly believe that restoring quality of life should be emphasised in guidelines as one of the most important therapeutic goals.

## 1. Introduction

Inflammatory bowel disease (IBD) is a chronic, relapsing inflammatory condition of the gastrointestinal tract, encompassing Crohn’s disease (CD), ulcerative colitis (UC), and indeterminate (unclassified) colitis. Although its exact aetiology remains unclear, IBD is believed to result from a dysregulated immune response in genetically predisposed individuals, triggered by environmental and microbial factors [1,2]. The disease course is often unpredictable, with relapsing gastrointestinal inflammation episodes that sometimes cause dramatic symptoms. The most common among these are diarrhoea, bloody stools, and bowel urgency in UC, and abdominal pain and diarrhoea in CD. Disease relapses can occasionally present with fever, fatigue, weight loss, and the loss of appetite [1]. While primarily impacting the gastrointestinal (GI) tract, IBDs may also manifest in other organ systems in approximately one-third of patients. Extraintestinal manifestations (EIMs) may present during the active or quiescent phases of IBD. They may occur before, at the same time, or after a diagnosis of inflammatory bowel disease has been made. Their response to treatment varies—some resolve with the control of intestinal inflammation, while others demand additional, organ-specific interventions. EIMs most frequently affect the musculoskeletal system (peripheral or axial arthritis and enthesitis), as well as the skin (pyoderma gangrenosum, erythema nodosum, and aphthous ulcers), the hepatobiliary system (notably primary sclerosing cholangitis), and the eyes (such as episcleritis, anterior uveitis, and iritis) [3,4]. A recent systematic review of the literature concluded that the prevalence of IBD in Europe is close to 0.3%, equivalent to around 2.25 million people [5,6].

The standard therapeutic approach to IBD involves using pharmacotherapy to control symptoms, mainly by modulating the immune response. These include aminosalicylates, corticosteroids, immunomodulators, and, more recently, biologics (anti-TNFs, ustekinumab, vedolizumab, mirikizumab, risankizumab, guselkumab) and small molecules (tofacitinib, upadacitinib, filgotinib, ozanimod, etrasimod). Despite the recent progress in conservative treatment in IBD, surgery continues to play a significant role in the overall therapeutic approach. Recent studies have shown that the surgical rate for Crohn’s disease declined from 10% to 8.8% (*p* < 0.001), while for ulcerative colitis, it decreased from 7.7% to 7.5% [1,7]. Croatia is currently among the countries providing its patients with the advanced, modern therapeutic options outlined above. The Sestre Milosrdnice University Hospital Centre has its own team of IBD specialists, tasked with personalising therapeutic interventions to the individual needs of each patient.

IBD is characterised by periods of relapse and remission, with symptoms such as diarrhoea, rectal bleeding, urgency, faecal incontinence, and abdominal pain significantly impacting the patients’ quality of life, which, compared to healthy individuals, is reduced in these patients [8,9,10,11,12]. The World Health Organisation defines quality of life as an individual’s perception of their position in life in the context of the culture and value systems in which they live, and about their goals, expectations, standards, and concerns [13]. The quality of life of the patients with IBD is often assessed using questionnaires, such as the generic EUROHISQoL [14] and the IBD-specific IBDQ [15]. Earlier research and guidelines have not strongly emphasised improved quality of life as a therapeutic goal, but this perspective has changed significantly in the last decade. Since 2013, a growing number of unique publications have been released each year, focusing on quality of life in patients with IBD. In the recent period, patient-reported outcomes and quality of life have grown in importance, both in research and in clinical trials [10,11,16]. The recent CONFIDE study [8] aimed to define the difference in perspectives between patients and healthcare providers regarding symptom severity and impact. Among other shortcomings in communication, they found a key discrepancy in the importance given to bowel urgency and the fears associated with this symptom.

The following question arises: which disease phenotype has a greater impact on the quality of life in patients with IBD? It was previously believed that Crohn’s disease (CD) had a greater impact on quality of life than ulcerative colitis (UC), which is reflected in the literature by a significantly higher volume of research focused on CD. However, when you look at the results of this study, you can see that the impact of CD and UC on quality of life is similar [17].

Considering the newfound importance of quality of life (QoL) and its novelty, this cross-sectional study was conducted to assess the following: (1) the impact IBD and specific therapy (conventional vs. advanced options) have on various aspects of patients’ QoL; (2) the patients’ perception of IBD medications and possible surgical interventions; and (3) the perception of QoL as a therapeutic goal.

## 2. Materials and Methods

### 2.1. Participants

The Sestre Milosrdnice University Hospital Centre (UHC) is a tertiary IBD centre that cares for over 1300 IBD patients. From 10 October 2022 to 21 February 2023, all patients who had a medical appointment were offered two questionnaires to fill out. The inclusion criteria were age ≥ 18 years and a confirmed diagnosis of UC or CD, as determined by a combination of clinical, biochemical, endoscopic, and histological examinations. Each patient was required to complete the anonymous questionnaire one time only. We aimed to have half of our respondents be bio-naïve patients, on conventional treatment options, and the other half on advanced therapeutics (biologics, small molecules). Patients who underwent surgery at any point in their treatment were excluded from this study. Both questionnaires were anonymous. Upon completing the questionnaires, all patients were referred to a psychiatrist integrated into the hospital’s multidisciplinary IBD team. The Ethics Committee of Sestre Milosrdnice UHC approved this study (Approval code: 251-29-11-26-01-8) on 11 February 2021.

### 2.2. Survey

The survey was conducted over four months to align with the median interval between follow-up visits in our IBD cohort, thereby ensuring a representative sample of the overall patient population.

Our first questionnaire was composed of two leading questionnaires for assessing quality of life, the IBDQ-32 [11,13,18] and the Work Productivity and Activity Impairment Questionnaire (WPAI) [19]. They consisted of several sets of questions that examined the patient’s perception of their health, the impact of the disease on their emotions, social interactions, and physical activities, as well as fatigue-induced disturbances and limitations in daily life. All questions had two or more provided answers.

The second questionnaire represented the IBD Disk, a simple tool designed to assess how ten different aspects of the disease affected patients’ daily lives [20]. The patients were asked to choose a number from 1 to 10 (1 = no impact; 10 = significant impact) to describe how each aspect impacted their everyday lives. The results were then shown on a simple and colourful visual scale. By completing it at each visit, patients and doctors had an immediate visual representation of the disease-caused disability, allowing its progress over time to be monitored.

### 2.3. Statistical Analysis

Descriptive statistical methods were used in the analysis, and the normal distribution was assessed using Kolmogorov–Smirnov and Shapiro–Wilk tests. Continuous variables were presented as medians with corresponding interquartile ranges or means with standard deviations, depending on the results of the previous tests. Categorical variables were shown as valid percentages and the absolute numbers of participants. Differences between categories (CD vs. UC respondents and patients treated with conventional vs. advanced therapies) were assessed using the Chi-square test and the Mann–Whitney U test.

Data analysis was performed using IBM SPSS Statistics, Version 30.0 (IBM Corp., Armonk, NY, USA). All the raw data are available as a Appendix A.

## 3. Results

The questionnaires were completed by a total of 159 participants, with a response rate of 91.6%. In total, 51% of the participants were males, 47.9% had UC, and 49.4% had been or were currently treated with biologics. The median age was 37.5, with an IQR of 25–54.

There was no statistically significant difference between the answers given by patients with CD as opposed to those with UC (Table 1 and Table 2). Hence, we decided to present their answers together, under the umbrella term of IBD.

### 3.1. First Questionnaire

Descriptive statistics for all questions in the first questionnaire can be found in Table 3, Table 4, Table 5 and Table 6. Questions oriented towards the perceived general health impression of the respondents are presented in Table 3. The results indicated that most respondents considered their health to be good, but only a few (12.8%) claimed, with absolute certainty, that their health was at the level of healthy individuals. Only 13 (8.3%) respondents claimed their health was excellent. Still, a significant number of respondents (48.1% and 52.6%, respectively) partially agreed with those two claims.

The disease impact on various forms of physical activity is presented in Table 4. The results indicated that our group of respondents felt significantly limited when performing strenuous physical activities, with 95 (60.1%) expressing at least minor limitations. Other lighter forms of physical activities were not affected as much by the disease.

The third group of questions (Table 5 and Table 6) was centred on the impact of disease-induced emotional problems, symptoms, and fatigue on our respondents’ everyday and social lives. Only two answers were given for these questions. The findings indicated that IBD impacted all these aspects of our respondents’ lives. Work-related issues were prominent, with 53 (33.8%) of the respondents believing they were professionally or physically limited due to health problems, and 30 (19.5%) feeling that they missed work more often because of their illness. Also, IBD proved itself to be a considerable emotional burden, with 77 (48.7%) of the respondents believing they were exposed to more stress than others, and 51 (32.3%) reported having anxious or depressive thoughts. One of the leading sources of anxiety was the anticipation of future symptoms, clearly shown by 74 (46.5%) respondents considering the locations and accessibility of restrooms when planning any outing.

Additionally, their current pharmacological therapy caused fear in 26.5% of respondents, and even more of them, as many as 96 (61.1%), experienced fear due to the possibility of a future surgical procedure. Still, 120 (94.5%) of the respondents believed biologics improved their quality of life. Lastly, respondents were asked about the impact of the disease on their lives during the remission phase. Altogether, 119 (75.3%) respondents believed that the disease affected their lives at least mildly during remission, and 20 (12.6%) respondents felt that the disease significantly affected their lives during remission.

### 3.2. Second Questionnaire

The results of the second questionnaire, the IBD Disk, are presented graphically in Figure 1. The values on the graph represent the medians of responses from all respondents (between 152 and 157 responded to each question). We could see that all aspects of our respondents’ lives were, at least to a mild degree, affected by their disease. The most significantly affected was energy, which was to be expected as fatigue is a common symptom of IBD.

### 3.3. Conventional vs. Advanced Therapeutic Options

Additional analyses were performed to examine whether differences exist in specific quality of life (QoL) domains between patients receiving conventional therapy (mesalazine, azathioprine, corticosteroids) and those treated with advanced therapies (infliximab, adalimumab, vedolizumab, ustekinumab, tofacitinib). The results of this analysis can be found in Table 7, Table 8 and Table 9. We found a statistically significant difference in only one question, namely the one referring to the impact of the disease during the periods of remission. There was a noticeable, but statistically insignificant, difference in the answers to the question about the amount of stress our patients were exposed to and the question assessing the importance of toilets during transportation.

## 4. Discussion

For decades, the conventional approach to medicine has given precedence to addressing the physical aspects of illness, with achieving clinical, biochemical, and endoscopic response and remission as the primary therapeutic objectives in IBD. In recent times, with the advancements in treatment options, an increased emphasis has been placed on treatment targets from the patient’s perspective and quality of life for patients with IBD.

Consistent with previous studies [10,11,12], our results showed that patients with IBD have a significantly reduced quality of life. A significant proportion of patients did not perceive themselves as being as healthy as others (21.8%) and disagreed with the statement that their overall health was excellent (28.2%). It is important to highlight that the decline in the quality of life of our patients occurred not only during active disease periods but also persisted during the periods of remission, as in previous research [11,21]. This data alone emphasised the importance of disease clearance, restoring quality of life as one of the main goals in treating IBD. Still, the single most important predictor of poor quality of life was disease activity [11,12,22,23,24,25]. In line with earlier research, our results showed that most variables affecting our patients’ quality of life were directly related to faecal urgency or faecal incontinence as the consequence of disease activity. For example, a significant portion of our patients stated that while planning any outing, they needed to consider the location and availability of toilets. They also preferred travel options with toilets, and while in public places, they often chose seats near toilets. Faecal urgency and faecal incontinence are distressing symptoms of IBD that lead many patients to avoid work, social, and physical activities. Even more concerning, earlier research showed that healthcare providers often severely underestimate these symptoms [8]. Defining therapeutic goals together is key in enhancing the overall quality of care and may also contribute to improved treatment adherence. Furthermore, our patients expressed concerns about the pharmacological therapy they were receiving, and many of them expressed fear due to possible surgical procedures in the future (26.5% and 61.1%, respectively). We find this information crucial, urging us physicians, together with our IBD nurses, to provide a more thorough explanation of all treatment options and to possibly alleviate any concerns they may have. This study did not find a statistically significant difference in the answers provided by conventional patients compared to those receiving advanced therapeutic options (biologics or small molecules), except in one instance. A larger percentage of patients receiving advanced therapeutic options felt their disease impacted their lives in periods of remission. This difference could be explained by the fact that a significant percentage of advanced treatment options required daily hospital administration, which consumed a considerable amount of time and reduced patients’ independence.

Inflammatory bowel disease is marked by frequent relapses that often require hospitalisation or staying at home. It is no surprise that some of our patients considered themselves to miss work more often than healthy individuals. They believed their employers were required to help them solve these issues, either by allowing them to work from home or offering flexible working hours. Previous research found similar results as, compared to the healthy population, IBD patients worked reduced hours more often, mainly due to a perceived lack of control over their symptoms [26]. One European study [27], including 4670 IBD patients, found that 40% of them had to adjust their work life due to illness, most often by switching to working from home or working reduced and flexible hours. The most prevalent reasons for absenteeism were fatigue, scheduled or emergency doctor’s appointments, and spasmodic abdominal pain. IBD’s negative impact on patients’ professional lives has shown a correlation with poorer quality of life in several studies [22,25,27,28]. Absenteeism, often coupled with high treatment costs, can present a significant financial burden for individuals and the countries they live in. Advancements in therapeutic options could not only mean better quality of life for patients but also significant savings for healthcare systems in their countries, regardless of the price of the drug itself [29,30]. Until then, educating the wider population about the challenges IBD patients face every day, together with advocating for systematic changes that allow them flexibility in work, may prove beneficial in the long run. Following this research, better cooperation was established between our institution and the Croatian Crohn’s and Ulcerative Colitis Association, a national association of patients with IBD, which has mechanisms to influence the improvement of working conditions for our patients.

The chronic nature of IBD, its prevalence in young people, and the need for long-term treatment present a burden these patients carry every day, which can negatively impact their mental state. Compared to the general population, research confirms a significantly higher frequency of anxiety and depression among patients with IBD, while defining their role as independent predictors of poorer quality of life [31,32,33] and IBD-related disability [34]. Authors Garcia-Alanis M. et al. [31] have, with the use of diagnostic interviews, found 56.7% of patients in their group, a percentage much larger than in studies on the general population, suffer from at least one of the major mental disorders. Furthermore, research indicates that depression and anxiety, while not capable of causing IBD, might still promote disease exacerbations, which we can notice in everyday practice. Psychological morbidity can induce abnormal abdominal pain perception, sleep dysfunction, negative illness perception, and non-adherence to medication [35]. The key takeaway from most studies was that examinations for psychological disturbances in IBD patients need to be conducted more often when taking a comprehensive approach to their treatment. Shortcomings in this area can be seen in the study by Engel K et al. [24] which showed that 80% of recently diagnosed IBD patients felt they did not receive sufficient psychological support from their physician. The same study also showed a positive correlation between a good patient–physician relationship and patients’ quality of life. Our findings aligned with prior research, as our patients perceived themselves as underachievers and reported absenteeism due to emotional issues. Many expressed heightened stress levels, attributing frustration to their illness. A considerable number of patients stated they have anxious or depressive thoughts, underscoring a perceived lack of understanding and support from the people around them. Of particular concern was that a significant number of our patients felt a lack of empathy from their friends, coworkers, and, what particularly surprised us, their family. As physicians, we also have an important role in taking care of our patients’ mental health. Tight cooperation between gastroenterologists and psychologists or psychiatrists can prove very important in managing this often-neglected aspect of IBD. Unfortunately, due to obligations and a lack of time, we sometimes overlook this aspect and, when monitoring patients, rely more on biochemical parameters and selected clinical symptoms than on the patient’s general condition. After this research at our clinical hospital centre, we addressed this by including a psychiatrist as a constitutive member of our IBD team and making it a standard of care to refer every newly diagnosed IBD patient to them.

Fatigue, mental health issues, and malnutrition with a loss of muscle mass are all consequences of the chronic nature of IBD that can limit patients’ physical abilities. Only a few of our patients felt like their disease limited them from performing moderate activities such as climbing, walking long distances, or squatting. In contrast, a significant number of patients felt limited when performing intense activities. This followed the same results in previous research that found patients with IBD were less active than the healthy population, mainly due to symptoms like abdominal or joint pain or the fear of their onset [36,37]. Patients currently in severe, active phases of IBD are often in hospitals and thus cannot be physically active. However, individuals in remission or experiencing milder forms of IBD could benefit from physical activity, given its established role in complementing the treatment of other chronic diseases [38]. The possible positive effects of exercise include improved physical and mental health, better nutritional status, higher bone density, and the restoration of muscle strength, all of which could lead to better quality of life. Although studies have shown that individuals with IBD tolerate physical activity, and that it can positively impact their quality of life, the ideal type of training for this population is not known [36,39]. For this reason, authors Erp et al. [37] conducted a study with 25 patients with IBD in remission. All of them were invited to participate in a 12-week workout program specifically designed for them by a sports medicine specialist and under the constant surveillance of a physical therapist. Following 12 weeks of this program, participants showed a significant reduction in body fat percentage, enhancement in muscle strength, improved cardiovascular health, better results in quality-of-life questionnaires, and fatigue levels equal to those in the healthy population. To conclude, individuals with IBD have a significant benefit from targeted physical activity. Collaboration between their physicians and physiotherapists, or at the very least the availability of personalised programs, would positively impact their treatment and overall well-being. For instance, at the onset of an IBD diagnosis, a gastroenterologist should explain the importance of physical activity, alleviate anticipatory anxiety IBD patients might have regarding their symptoms and exercise, and refer them to a physiotherapist specialised in IBD patients, with a workout plan specifically tailored to their needs. At the same time, offering a nutritionist-approved diet plan may even further improve adherence.

This questionnaire has improved everyday clinical practice in our hospital by making us think about the patients’ quality of life even more, which is the first step in improving it. Communication with patients is vital for improving the quality of IBD treatment. Even when pressured by daily obligations, we cannot ignore quality of life when monitoring our patients, especially questions about faecal urgency and faecal incontinence, which are tremendously important for the quality of life of our patients. Our centre’s experience has demonstrated that IBD nurses play a crucial role in the team, significantly enhancing communication and, consequently, patient care. A multidisciplinary IBD team, including both a psychiatrist as a standard member and a psychologist (on demand), should be the standard approach. As we have shown in our patients, IBD Disk can also offer practical and fast insight into the patient’s general condition. Therefore, it can be provided to patients at set intervals to objectively evaluate their perception of their illness activity, and this can be performed entirely by an IBD nurse.

This study had several limitations. It was conducted in a single centre, the Sestre Milosrdnice Clinical Hospital Centre, and as such was limited to a smaller pool of patients and their relative lack of diversity. Furthermore, we chose to include patients treated with different therapeutic options but did not differentiate between them. We excluded all patients who underwent surgery; thus, this large group of patients was not represented in this study. Additionally, we were surprised by the large number of patients who felt misunderstood by their families. We believe more research needs to be undertaken to determine how different societal circumstances (e.g., family dynamics, marital status, economic status, type of occupation) impact quality of life. Lastly, sleep is a vital determinant of overall quality of life that was not adequately covered in this study. Since mental disorders that impact sleep are more prevalent in this group of patients, we believe this to be an important area for further research.

## 5. Conclusions

Our findings, along with other studies, showed that IBD patients have reduced quality of life, not only in the periods of active disease but also during clinical remission. Therefore, quality of life must be considered a primary therapeutic goal for these patients. Physical symptoms were only partly responsible for diminishing quality of life, and other factors, such as mental health issues, were also found to impact quality of life. The development of new, efficient therapeutic modalities with better outcomes for IBD patients forces us to take a new approach to our patients. As physicians, we have an important role in taking care of our patients’ quality of life, but we often do not ask very simple questions. The increased awareness of the importance of quality of life is evident from a growing number of studies covering it, which is promising for IBD patients. We firmly believe that restoring quality of life should be emphasised in guidelines as one of the most important therapeutic goals. Only with a comprehensive personalised therapeutic approach, by restoring quality of life to the state it was before the onset of the disease, can we achieve the primary treatment target [40].

## Figures and Tables

**Figure 1 healthcare-13-01681-f001:**
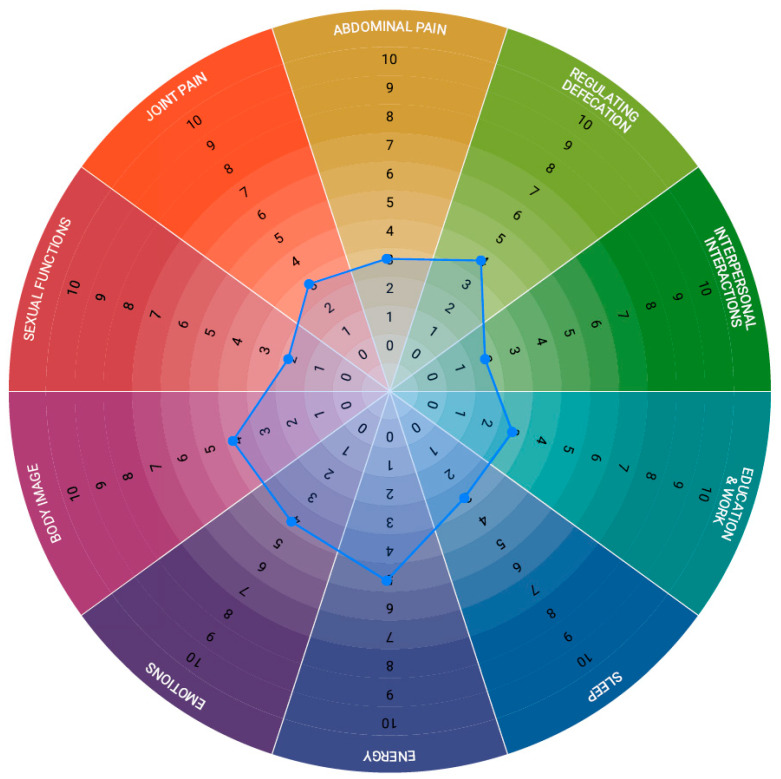
IBD Disk. This graph visually presents how much certain disease characteristics impact patients’ daily lives (0 = no impact, 10 = significant impact). Values shown on the graph represent the median values taken from all responses.

**Table 1 healthcare-13-01681-t001:** Comparison of IBD Disk responses between patients with UC and CD.

	Abdominal Pain	Regulating Defecation	Interpersonal Interactions	Education and Work	Sleep	Energy	Emotions	Body Image	Sexual Function	Joint Pain
Mann–Whitney U test	515.5	573.5	580.5	580.0	583.0	618.5	560.5	630.0	574.5	628.5
*p* value	0.183	0.520	0.554	0.694	0.578	0.741	0.321	0.842	0.632	0.825

**Table 2 healthcare-13-01681-t002:** Comparison of responses between patients with UC and CD.

Question	Mann–Whitney U Test	*p* Value
I believe I am more prone to getting sick than others.	568.5	0.261
I am equally as healthy as others.	528.5	0.105
I believe my condition will worsen.	611.5	0.524
My health is excellent.	627.5	0.652
When in remission, how much does your illness impact your life?	505.5	0.093
My condition prevents me from engaging in intense activities.	628.5	0.663
My condition prevents me from engaging in mild activities.	621.5	0.470
My condition prevents me from lifting and carrying bags.	637.0	0.641
My condition prevents me from climbing to the second floor continuously.	604.5	0.329
My condition prevents me from climbing the first floor continuously.	598.5	0.123
My condition prevents me from squatting, kneeling, or standing still.	525.0	0.065
My condition prevents me from walking for more than 1 km continuously.	632.5	0.610
My condition prevents me from walking for more than 100 m continuously.	650.0	0.630

**Table 3 healthcare-13-01681-t003:** Descriptive statistics for questions oriented towards perceived general health impressions.

Question	1, n (%)	2, n (%)	3, n (%)	4, n (%)
I believe I am more prone to getting sick than others.	16 (10.1)	39 (24.7)	33 (20.9)	70 (44.3)
I am equally as healthy as others.	20 (12.8)	75 (48.1)	27 (17.3)	34 (21.8)
I believe my condition will worsen.	9 (5.7)	29 (18.2)	64 (40.8)	55 (35)
My health is excellent.	13 (8.3)	82 (52.6)	17 (10.9)	44 (28.2)

1 = True; 2 = Partially true; 3 = I don’t know; 4 = Not true.

**Table 4 healthcare-13-01681-t004:** Descriptive statistics for questions assessing disease impact on physical activity.

Question	1, n (%)	2, n (%)	3, n (%)
My condition prevents me from engaging in intense activities.	27 (17.1)	68 (43)	63 (39.9)
My condition prevents me from engaging in mild activities.	5 (3.2)	28 (17.7)	125 (79.1)
My condition prevents me from lifting and carrying bags.	3 (1.9)	22 (14.1)	131 (84)
My condition prevents me from climbing to the second floor continuously.	3 (1.9)	32 (20.5)	121 (77.6)
My condition prevents me from climbing the first floor continuously.	2 (1.3)	12 (7.6)	143 (91.1)
My condition prevents me from squatting, kneeling, or standing still.	5 (3.2)	45 (28.8)	106 (67.9)
My condition prevents me from walking for more than 1 km continuously.	7 (4.4)	29 (18.4)	122 (77.2)
My condition prevents me from walking for more than 100 m continuously.	3 (1.9)	9 (5.7)	145 (92.4)
My condition prevents me from bathing or getting dressed.	1 (0.6)	12 (7.6)	145 (91.8)

1 = Yes, it significantly limits me; 2 = Yes, it slightly limits me; 3 = No, it doesn’t limit me.

**Table 5 healthcare-13-01681-t005:** Descriptive statistics for questions assessing various disease-induced everyday difficulties.

Question	YES, n (%)	NO, n (%)	Total
Due to health issues, I spent less time at work/other activities.	42 (26.8)	115 (73.2)	157
Due to health issues, I achieved less in the last month.	45 (28.5)	113 (71.5)	158
Due to health issues, I am restricted in terms of work/physical activities.	53 (33.8)	104 (66.2)	157
Due to health issues, I had to put in more effort to achieve the same goals.	67 (42.4)	91 (57.6)	158
Due to emotional issues, I spent less time at work.	34 (21.5)	124 (78.5)	158
Due to emotional issues, I achieved less than desired.	43 (27.2)	115 (72.8)	158
Due to emotional issues, I felt limited in work/physical activities.	33 (21.2)	123 (78.8)	156
I consider myself to be exposed to more stress than others.	77 (48.7)	81 (51.3)	158
I consider it difficult, or impossible, to lead a normal life.	20 (12.6)	139 (87.4)	159
Do you feel displeasure/shame because of your illness?	23 (14.6)	135 (84.9)	158
Does your illness present a source of frustration for you?	56 (35.4)	102 (64.6)	158
Do you consider that you have depressive/anxious thoughts?	51 (32.3)	107 (67.7)	158
Do you feel as though your family doesn’t understand you due to your illness?	24 (15.1)	135 (84.9)	159
Do you feel as though your friends don’t understand you due to your illness?	50 (31.6)	108 (68.4)	158
Do you feel as though your work colleagues don’t understand you due to your illness?	47 (29.6)	109 (69.9)	156
Do you feel stress/fear due to the unavailability of toilets?	69 (43.4)	88 (55.3)	157
Do you feel afraid due to the medications you are taking?	41 (26.5)	114 (73.5)	155
Do you feel fear due to the possibility of surgical procedures in the future?	96 (61.1)	61 (38.9)	157
Do you think you ever ruined an important moment due to your illness (cancelled or postponed an event)?	40 (25.2)	117 (74.5)	157
Do you feel like you often miss work due to your illness?	30 (19.5)	124 (80.5)	154
Do you feel that, due to your illness, you are less productive at your workplace (quiet, inactive during meetings)?	25 (16.1)	130 (83.9)	155
Do you feel like you are less motivated or increasingly agitated due to your illness?	61 (38.9)	96 (61.1)	157
Do you believe your employer should assist in addressing work-related challenges (work from home, flexible work hours…)?	76 (49)	79 (51)	155
Do you occupy seats closer to the restroom when going to bars, cinemas, restaurants, or other public places?	55 (35.9)	100 (64.1)	156
When planning any outing, do you consider the location of and availability of restrooms?	74 (46.5)	85 (53.5)	159
Do you prefer transportation options that have a toilet?	68 (42.8)	91 (57.2)	159
Do you believe biologics can improve your quality of life?	120 (94.5)	7 (5.5)	127

**Table 6 healthcare-13-01681-t006:** Descriptive statistics for the question assessing disease impact on life when in remission.

Question	1, n (%)	2, n (%)	3, n (%)	4, n (%)
When in remission, how much does your illness impact your life?	20 (12.7)	34 (21.4)	65 (41.1)	39 (24.7)

1 = Significant impact; 2 = Some impact; 3 = Mild impact; 4 = No impact.

**Table 7 healthcare-13-01681-t007:** Comparison of patient-reported outcomes between conventional and advanced treatment groups (questions with more than two answers).

	Conventional Therapy	Advanced Therapy	*p*-Value
I believe I am more prone to getting sick than others.	3 (2–4)	3 (2–4)	0.760
I am equally as healthy as others.	2 (2–3)	2 (2–3)	0.104
I believe my condition will worsen.	3 (3–4)	3 (2–4)	0.604
My health is excellent.	2 (2–4)	2 (2–3)	0.139
My condition prevents me from engaging in intense activities.	2 (2–3)	2 (2–3)	0.240
My condition prevents me from engaging in mild activities.	3 (3–3)	3 (3–3)	0.326
My condition prevents me from lifting and carrying bags.	3 (3–3)	3 (3–3)	0.355
My condition prevents me from climbing to the second floor continuously.	3 (3–3)	3 (3–3)	0.920
My condition prevents me from climbing the first floor continuously.	3 (3–3)	3 (3–3)	0.617
My condition prevents me from squatting, kneeling, or standing still.	3 (2–3)	3 (2–3)	0.167
My condition prevents me from walking for more than 1 km continuously.	3 (3–3)	3 (2–3)	0.209
My condition prevents me from walking for more than 100 m continuously.	3 (3–3)	3 (3–3)	0.074
My condition prevents me from bathing or getting dressed.	3 (3–3)	3 (3–3)	0.737
When in remission, how much does your illness impact your life?	2 (2–3)	2 (2.25–4)	0.036

The data are represented as medians of the attributed scale.

**Table 8 healthcare-13-01681-t008:** Comparison of patient-reported outcomes between conventional and advanced treatment groups (yes or no questions).

	Conventional Therapy	Advanced Therapy	*p*-Value
Due to health issues, I spent less time at work/other activities.	29.5%	23.7%	0.415
Due to health issues, I achieved less in the last month.	30.8%	27.3%	0.632
Due to health issues, I am restricted in terms of work/physical activities.	36.4%	32.5%	0.611
Due to health issues, I had to put in more effort to achieve the same goals.	41%	44.2%	0.694
Due to emotional issues, I spent less time at work.	20.5%	23.4%	0.667
Due to emotional issues, I achieved less than desired.	25.6%	29.9%	0.557
Due to emotional issues, I felt limited in work/physical activities.	21.1%	22.1%	0.877
I consider myself to be exposed to more stress than others.	41%	57.1%	0.045
I consider it difficult, or impossible, to lead a normal life.	16.5%	9.1%	0.169
Do you feel displeasure/shame because of your illness?	11.4%	18.4%	0.218
Does your illness present a source of frustration for you?	37.2%	33.8%	0.657
Do you consider that you have depressive/anxious thoughts?	35.4%	31.2%	0.571
Do you feel as though your family doesn’t understand you due to your illness?	15.2%	15.6%	0.946
Do you feel as though your friends don’t understand you due to your illness?	28.2%	36.4%	0.277
Do you feel as though your work colleagues don’t understand you due to your illness?	27.3%	34.2%	0.352
Do you feel stress/fear due to the unavailability of toilets?	49.4%	40%	0.194
Do you feel afraid due to the medications you are taking?	25%	27.6%	0.713
Do you feel fear due to the possibility of surgical procedures in the future?	67.5%	54.5%	0.098
Do you think you ever ruined an important moment due to your illness (cancelled or postponed an event)?	26%	24.7%	0.853
Do you feel like you often miss work due to your illness?	17.1%	22.7%	0.392
Do you feel that, due to your illness, you are less productive at your workplace (quiet, inactive during meetings)?	17.1%	14.5%	0.656
Do you feel like you are less motivated or increasingly agitated due to your illness?	44.2%	33.8%	0.186
Do you believe your employer should assist in addressing work-related challenges (work from home, flexible work hours…)?	43.4%	53.9%	0.194
Do you occupy seats closer to the restroom when going to bars, cinemas, restaurants, or other public places?	40.3%	31.6%	0.263
When planning any outing, do you consider the location of and availability of restrooms?	53.2%	39%	0.075
Do you prefer transportation options that have a toilet?	49.4%	33.8%	0.048
Do you believe biologics can improve your quality of life?	90.2%	97.3%	0.09

The data are represented as percentages of the positive answers.

**Table 9 healthcare-13-01681-t009:** Comparison of patient-reported outcomes between conventional and advanced treatment groups (IBD Disk).

	Conventional Therapy	Advanced Therapy	*p*-Value
Abdominal pain	3 (2–6)	3.5 (1–5.75)	0.988
Regulating defecation	5 (2–7)	4 (2–8)	0.978
Interpersonal interactions	3 (1–5)	2 (1–4)	0.269
Education and work	3 (1–5)	3 (1–5)	0.627
Sleep	2 (1–5)	3 (1–5)	0.788
Energy	5 (2–6)	5 (2–7)	0.984
Emotions	4 (2–6)	4 (1–6.75)	0.708
Body image	4 (1–5)	3 (1–7)	0.62
Sexual function	2 (1–5)	1 (1–5)	0.12
Joint pain	3 (1–6)	3 (1–7)	0.993

## Data Availability

The original contributions presented in this study are included in the article. Further inquiries can be directed to the corresponding author.

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
