# Peer review of "Impaired Quality of Life in Croatian IBD Patients in the Era of Advanced Treatment Options"

_healthcare, 2025, doi:10.3390/healthcare13141681_

Round 1
Reviewer 1 Report
Comments and Suggestions for Authors
There is no doubt that this is a good scientific manuscript in relation to Croatian IBD patients. and as such adds to the body of knowledge relating to IBD in different countries.
Generally the manuscript is reasonably well written. The cohort of patients surveyed is appropriate (and necessary) and, again, from a general point of view, the survey questions are appropriate and well organized.
Unfortunately the study parameters should have been researched in a more rigorous manner before it was conducted; the foregoing statement-now- means that some of the corrections necessary or advisable cannot, or may be difficult to, be undertaken.
Major problems include the following.
(1) IBD is not a single disease. It's an umbrella term for two main, chronic conditions: Crohn's disease and ulcerative colitis; the authors are well aware of this fact but do not attempot to separate the two conditions.
(2) Somewhat oddly, the authors include the term " advanced treatment operations" in the title of the manuscript but do not "knit/include/refer to" such concepts in the body of the manuscript in terms of their survey, results or for the benifit of the reader.
(3) In my experience the family is important in relation to the well-being of IBD patients, especially older patients. Yet, a whole host of questions-pertaining to close or wider (extended) family - members are not included; the foregoing would have substantially improved the manuscript.
The authors are invited to provide a scientific riposte to the three point raised above.
Author Response
Comment 1: (3) In my experience the family is important in relation to the well-being of IBD patients, especially older patients. Yet, a whole host of questions-pertaining to close or wider (extended) family - members are not included; the foregoing would have substantially improved the manuscript.
Response 1: Respected reviewer, thank you very much for taking the time to review our manuscript. As you have well observed, some advised changes would be very difficult to make. Regarding your third comment, we agree that including only one question pertaining to family and its role in IBD patients' lives is not enough. When choosing the questions for our questionnaire, we did not expect to find such a significant amount of our patients feel misunderstood by their families. Families have a crucial role in the lives of patients with chronic diseases such as IBD. This is an important area for future research, we hope mentioning this in the last paragraph of the Discussion section will be adequate for now.
Comment 2: (1) IBD is not a single disease. It's an umbrella term for two main, chronic conditions: Crohn's disease and ulcerative colitis; the authors are well aware of this fact but do not attempt to separate the two conditions.
Response 2: As you have stated, IBD is not a single disease, but rather a term mainly pertaining to two separate conditions: Crohn's disease and ulcerative colitis. Even though these diseases generally have important clinical differences, our study did not find a statistically significant difference in the impact the diseases have on their respective patients’ lives. Because of this, we decided to refer to both conditions collectively using the same umbrella term of IBD. We agree this information is important to disclose, so we have added it to the beginning of the Results section and explained further in the second-to-last paragraph of the Introduction section.
Comment 3: (2) Somewhat oddly, the authors include the term " advanced treatment operations" in the title of the manuscript but do not "knit/include/refer to" such concepts in the body of the manuscript in terms of their survey, results or for the benefit of the reader.
Response 3: The term “era of advanced treatment options” mentioned in the title was meant in a general context. Currently, Croatia is among the countries offering all advanced, modern therapeutic options (biologics, small molecules). University hospital center Sestre milosrdnice has its own IBD team of specialists, tasked with personalising therapeutic interventions to the individual needs of each patient. Our current treatment approach primarily involves anti-TNF agents as first-line therapy; however, this strategy may be adjusted based on individual patient needs and potential benefit from alternative agents. We disclosed this information in the second paragraph of the Introduction section.
Reviewer 2 Report
Comments and Suggestions for Authors
The author is recommended to further detail the methodology used, including study design, sample size and inclusion/exclusion criteria, to strengthen reproducibility and transparency.
Including a more robust statistical analysis and justifying the choice of tests used may improve the robustness of the results and their interpretation.
It is important that the author explicitly addresses the limitations of the study and suggests areas for future research, which can enhance the credibility and depth of the work.
The author should decrease the level of plagiarism which is 86% overall with 37% minor changes and 17% identical to other papers.
Author Response
Comment 1: The author is recommended to further detail the methodology used, including study design, sample size and inclusion/exclusion criteria, to strengthen reproducibility and transparency.
Response 1: Respected reviewer, thank you very much for taking the time to review our manuscript. As outlined in the Introduction, this was a cross-sectional study conducted over a 4-month period. During this time, we randomly and consecutively invited patients attending our outpatient clinic to complete the questionnaire. The 4-month timeframe was purposefully chosen to align with the median interval between follow-up appointments in our IBD cohort, thereby ensuring that the sample would be representative of the entire patient population (added to the manuscript section Survey). The inclusion and exclusion criteria are clearly detailed within the manuscript.
Comment 2: Including a more robust statistical analysis and justifying the choice of tests used may improve the robustness of the results and their interpretation.
Response 2: Thank you for your comment. We have expanded the Statistical Analyses section accordingly.
Comment 3: It is important that the author explicitly addresses the limitations of the study and suggests areas for future research, which can enhance the credibility and depth of the work.
Response 3: We wholeheartedly agree with this comment and, accordingly, made sure to discuss the limitations of our study and suggest areas for future research in the last paragraph of the Discussion section.
Comment 4: The author should decrease the level of plagiarism which is 86% overall with 37% minor changes and 17% identical to other papers.
Response 4: Thank you for this observation. Certain phrases and sentences pertaining to IBD, its aetiology, epidemiology, clinical findings and even impact on quality of life have been stated many times before, and we may have accidentally repeated them. We took the time to rewrite certain parts of the Introduction, Discussion and Conclusion sections. We hope you will find this new version adequate.
Reviewer 3 Report
Comments and Suggestions for Authors
The peer-reviewed study aims to assess how the manifestations of IBD affect the lives of patients and how IBD patients perceive their disease.
Inflammatory bowel diseases affect a huge number of patients worldwide. The worst thing is that the total number of patients is constantly increasing, which affects many young people. The reason is probably an unhealthy lifestyle - lack of exercise and improper diet. That is why this study is interesting and very relevant. The good thing is that the patients studied are half men and half women, which objectively shows the impact of the disease on both sexes.
The questionnaire contains interesting questions that are asked precisely.
The results obtained are very well illustrated.
I recommend that the introduction be expanded and the following points be discussed in more detail:
1. Symptoms and manifestations of IBD that worsen the quality of life
2. Conventional drug treatment - advantages and disadvantages, side effects with examples
3. Unconventional treatment and therapy with natural products - advantages and limitations
Question to the authors - has the individual condition of patients during sleep been studied, and do complaints of IBD affect the sleep of patients? Is this related to mental disorders, because your data show that 56.7% of patients in their group suffer from at least one of the main mental disorders that affects their sleep, but how? The authors have not commented in detail only on the impact of IBD on sleep, falling asleep, and sleep quality. Do you have any studies on this issue?
There are many positive results from this study that would be good to offer to other clinics that treat patients with IBD, including increasing physical activity, psychological support, and training. That is why I recommend publishing this article, but after including the additional information I offer.
Author Response
Comment 1: 1. Symptoms and manifestations of IBD that worsen the quality of life
Response 1: Respected reviewer, thank you very much for taking the time to review our manuscript. Defining different aspects of quality of life is more important than previously perceived. Studies such as DEFINE highlight how patients and their physicians often hold differing views when defining the most important therapeutic goals. We can see the need for improvement in this area and, accordingly, took the time to expand on it further in the first paragraph of the Introduction section. We hope you will find it adequate.
Comment 2: 2. Conventional drug treatment - advantages and disadvantages, side effects with examples
Response 2: Thank you for another constructive proposal. We made sure to discuss the conventional drug options in more detail in the second paragraph of the Introduction section.
Comment 3: 3. Unconventional treatment and therapy with natural products - advantages and limitations
Response 3: Although today we have many therapeutic options when treating IBD, it is true that up to one third of patients have difficulties maintaining complete remission. Keeping this in mind, using anything we have at our disposal, conventional or unconventional, that can help our patients is crucial. In our Clinical hospital center, when first diagnosed with IBD, each patient receives a brochure titled “Supportive, additional and alternative treatment in inflammatory bowel disease”, aimed at further elaborating exactly on this. If it would be of value to you, we can send a translated version for further reading. In the interest of preserving the current format, we believe that incorporating this information into our manuscript at this stage would be challenging.
Comment 4: Question to the authors - has the individual condition of patients during sleep been studied, and do complaints of IBD affect the sleep of patients? Is this related to mental disorders, because your data show that 56.7% of patients in their group suffer from at least one of the main mental disorders that affects their sleep, but how? The authors have not commented in detail only on the impact of IBD on sleep, falling asleep, and sleep quality. Do you have any studies on this issue?
Response 4: This is a significant observation. We agree that sleep disorders play a vital role in quality of life. Unfortunately, we did not include questions about sleep quality in our questionnaire, but rather, we only cited authors who made the correlation between mental disorders and sleep dysfunction. Because of this, we highlighted sleep as an important area for potential future research in the last paragraph of the Discussion section.
Reviewer 4 Report
Comments and Suggestions for Authors
- More detailed definition of IBD is needed at the beginning of the Introduction.
- There is also a third subtype, named indeterminate colitis.
- The currently used therapeutical options should be at least in general detailed.
- It is also not detailed how IBD would affect quality of life - sleep, etc is not specific enough.
- "Older research" - please rephrase.
- "Most of the respondents were patients undergoing biological therapy." - This is a results, not a method, and please specify with numbers.
- "and a confirmed diagnosis of UC or CD based on clinical, endoscopic, and histological examinations" - The final decider is histology, so those patients who does not have histologically confirmed IBD, should be exlcuded.
- Ethical approval number and date are not written.
- So even the initiated questionnaires were not originally made by the authors. They sentences listed are highly subjective.
- There is no detailed description of the examined cohort. Since when they had IBD? What kind of treatment were they given? Did they ever undergo surgery? Were psychologic/psychiatric help given to them?
- In a questionnaire like this, sociological factors should be considered, as well. Do these patients have families, who can help them? Are they married? What kind of work do they do?
- Discussion do not summarize the literature data the study isi based upon, even though in the introduction, it is mentioned that hundreds of papers already exist.
- "Consistent with previous studies, our results show that patients with IBD have a significantly reduced quality of life." - Please explain this with numbers.
- The Discussion often does not include numbers, just general sentences.
- Please make the discussion section more transparent and organised.
- The possible solutions the authors provide for these people are highly superficial. If you are a gastroenterologist, and work with these patients, you should come up with a plan then. For example: At the first IBD diagnosis, automatic psychologic consultation for all patients. Make the patients fill out the questionnaire at certain periods, or when they are relapsing, etc, and then take further actions.
- There is no list of abbreviations.
- The formatting on the References has many mistakes.
Author Response
Comments 1-5: 1. More detailed definition of IBD is needed at the beginning of the introduction.
- There is also a third subtype, named indeterminate colitis.
- The currently used therapeutical options should be at least in general detailed.
- It is also not detailed how IBD would affect quality of life – sleep, etc is not specific enough
- “Older research” – please rephrase
Response 1-5: Respected reviewer, thank you very much for taking the time to review our manuscript. You brought some very important observations, highlighting the need to strengthen and improve our Introduction section. Each of your comments has now been addressed in the revised Introduction. We hope you will find it adequate.
Comment 6: “Most of the respondents were patients undergoing biological therapy” – This is a results, not a method, and please specify with numbers.
Response 6: This is another valuable input. The sentence was removed from the Methods section. The exact percentage of patients undergoing biological therapy is 49,4%. We have already disclosed this in the first paragraph of the Results section.
Comment 7: “and a confirmed diagnosis of UC or CD based on clinical, endoscopic, and histological examinations” – The final decider is histology, so those patients who does not have histologically confirmed IBD, should be excluded.
Response 7: The sentence referenced here (found in 2.1. Participants) could be refined for clarity. A confirmed diagnosis of UC or CD is based on a combination of clinical, endoscopic and histological examinations, with histology as the final decider. In our centre, every patient undergoing endoscopic evaluation for IBD is biopsied for further histological confirmation.
Comment 8: Ethical approval number and date are not written.
Response 8: Ethical approval number and date were written in the Institutional Review Board Statement section at the end of the manuscript. We have now added both to the first paragraph of the Materials and methods section as well. Thank you for this remark.
Comment 9: So even the initiated questionnaires were not originally made by the authors. They sentences listed are highly subjective.
Response 9: The two questionnaires used in this study were based on existing validated questionnaires, while trying to minimize subjectivity. We thank you for your feedback, and will surely keep it in mind when conducting any future research. Unfortunately, it is impossible to address these issues in this study.
Comment 10: There is no detailed description of the examined cohort. Since when they had IBD? What kind of treatment were they given? Did they ever undergo surgery? Were psychologic/psychiatric help given to them?
Response 10: The feedback provided in this comment is sound, so we appreciate it. We did not ask our respondents how long they had had their IBD diagnosis. We know the mean disease length for patients treated in CHC Sestre milosrdnice, so we can assume this cohort is similar, as it represents an average sample of patients treated in our CHC, but we would not feel comfortable writing these assumptions in our manuscript. As stated in response 6, 49,4% of respondents were treated with biologics. We aimed to have half of our respondents on older treatment options and the other half on advanced options (biologics and small molecules). Patients who underwent surgery have been excluded from this study. After filling out this questionnaire, every patient was referred to a psychiatrist, who is a part of our hospital’s IBD team. All this is now disclosed in the revised version of section 2.1. Participants.
Comment 11: In a questionnaire like this, sociological factors should be considered, as well. Do these patients have families, who can help them? Are they married? What kind of work do they do?
Response 11: Of the total number of 160 respondents,159 have answered a question pertaining to their family's understanding of their disease. With this information, we assumed they had families. We agree that this is an important area for future research; therefore, we highlighted this in the last paragraph of the Discussion section, among other suggestions for future research.
Comment 12: Discussion do not summarize the literature data the study isi based upon, even though in the introduction, it is mentioned that hundreds of papers already exist.
Response 12: Thanks for that very constructive observation. We have made significant changes, and we hope that we have at least partially improved the Discussion section.
Comment 13: "Consistent with previous studies, our results show that patients with IBD have a significantly reduced quality of life." - Please explain this with numbers.
Response 13: We agree that the highlighted sentence was not explained enough in the context of our results. We made sure to be more precise this time.
Comment 14: The Discussion often does not include numbers, just general sentences.
Response 14: This is a good observation. The numbers should be kept in the Results section as much as possible. We made sure to update the Discussion as much as possible, while keeping this in mind.
Comment 15: Please make the discussion section more transparent and organised.
Response 15: While addressing your other comments and the comments of other reviewrs we revised significant parts of this section. While doing so, we made sure to review and organise it for an easier and better flow of reading. Thank you for highlighting this.
Comment 16: The possible solutions the authors provide for these people are highly superficial. If you are a gastroenterologist, and work with these patients, you should come up with a plan then. For example: At the first IBD diagnosis, automatic psychologic consultation for all patients. Make the patients fill out the questionnaire at certain periods, or when they are relapsing, etc, and then take further actions.
Response 16: The comment is sound, and we appreciate it. In the revised Discussion section, we made sure to offer a concrete solution to every major quality of life issue that IBD patients face. We also elaborated on specific solutions that have been established as standard practice at our Clinical Hospital Centre, along with our experiences implementing them. These comments can be found at the end of each paragraph of the Discussion section.
Comment 17: There is no list of abbreviations.
Response 17: We have now added the list of abbreviations at the end of the manuscript. Thank you for reminding us.
Comment 18: The formatting on the References has many mistakes.
Response 18: We made sure to present the references in Vancouver citation style. We tried our best to fix any mistakes. If any mistakes remain, we would be happy to improve wherever possible.
Round 2
Reviewer 1 Report
Comments and Suggestions for Authors
I thank the authors for taking into account my comments relating to the first version of the manuscript. Likewise, the authors are thanked for their riposte in relation to the foregoing.
The manuscript is now fit for purpose and publication can go ahead, with the approv al of the journal editors
Author Response
Thank you for your valuable comments that helped improve our manuscript.
Reviewer 2 Report
Comments and Suggestions for Authors
The author has adequately covered the requested review with respect to previous findings in his research work, therefore, acceptance of the manuscript is recommended. The author's attention to the recommendations favors the understanding of his work and its impact.
Author Response

(The authors gave the same response as above.)

Reviewer 3 Report
Comments and Suggestions for Authors
I thank the authors for accepting my recommendations and responding to my comments. The additional information they have added enriches the manuscript, and it looks and sounds much better. I have read the new data added in the paragraphs. I also agree with the authors' response regarding unconventional preparations, and I would like to familiarize myself with the brochure they give to their patients, as they suggested. I think the authors should continue their research, as they mentioned at the end, paying special attention to the sleep patterns of these patients.
I recommend the publication of the manuscript in this form.
Author Response

(The authors gave the same response as above.)

Reviewer 4 Report
Comments and Suggestions for Authors
- Table 1 - What do numbers next to Mann-Whitney U-test mean? Are they medians, if yes, to what? Or results of test statistics? If the latter, it is not necessary to present this way in the manuscript. Same applies to Table 2.
- Please rephrase the components of Table 1 - for example Energy and sleep is not informative.
- Table 2- some p values are written as .261, while others as 0.065 - Please unify!
- The results of statistical analysis is not represented in the abstract.
- Figure 1 represents that even though these patients are suffering from a serious, relapsing disease, their numbers are not so high, aka are they feeling/treated so well?
- Description of the second questionnaire is rather short.
- It is not clear what results is Table 1 based upon, while the same parameters are described at the second questionnaire section.
- We aimed to have half of our respondents be bio-naïve patients, on conventional treatment
options, and the other half on advanced therapeutic. - Why isn't study comparing these populations with the questionnaires?
Author Response
Comment 1: Table 1 - What do numbers next to Mann-Whitney U-test mean? Are they medians, if yes, to what? Or results of test statistics? If the latter, it is not necessary to present this way in the manuscript. Same applies to Table 2.
Response 1: The numbers next to the Mann-Whitney U-test represent test statistics. We fully agreed with you, but this addition was requested by another reviewer to our manuscript. Should you wish, we can exclude those elements and report the data per standard conventions.
Comment 2: Please rephrase the components of Table 1 - for example Energy and sleep is not informative.
Response 2: As stated in Table 1 description, these represent the comparison in responses to questions in IBD Disk (How much does your disease impact a certain part of your life?). For example, Energy represents answers to the question “How much does you disease impact your everyday energy?”. We hope this clarifies the confusion.
Comment 3: Table 2- some p values are written as .261, while others as 0.065 - Please unify!
Response 3: Thank you for pointing this out, the numbers have now been unified!
Comment 4: The results of statistical analysis is not represented in the abstract.
Response 4: We have added a sentence referring to our statistical analysis to the abstract. Thank you for this observation.
Comment 5: Figure 1 represents that even though these patients are suffering from a serious, relapsing disease, their numbers are not so high, aka are they feeling/treated so well?
Response 5: As stated in section 2.1. Participants, the included respondents were all patients suffering from IBD who had a medical appointment. These included both patients in full clinical remission and those suffering from milder symptoms. Patients with severe relapses are most often hospitalised, thus they are not represented in this study. We believed this to be self-explanatory, due to the relapsing nature of IBD.
Comment 6: Description of the second questionnaire is rather short.
Response 6: The description of the second questionnaire (3.2. Second questionnaire) has been expanded, as per request. Since the point of IBD Disk is visual representation of symptoms, we believed the fewer words used, the better.
Comment 7: It is not clear what results is Table 1 based upon, while the same parameters are described at the second questionnaire section.
Response 7: The results in Table 1 represent the statistical difference between the answers given by patients suffering from CD to those suffering from UC, using the Mann Whitney U test. As shown in Table 1, there was no statistical difference between the two groups for any questions. This stated at the beginning of section 3. Results.
Comment 8: We aimed to have half of our respondents be bio-naïve patients, on conventional treatment options, and the other half on advanced therapeutic. - Why isn't study comparing these populations with the questionnaires?
Response 8: Your line of thinking is sound. We have compared these populations but found a statistically significant difference in only one question. We have now added this information in a new section 3.3. Conventional vs advanced therapeutic options. We have further commented on these findings at the end of paragraph two of the Discussion section. We hope you will find this additional data and analysis satisfying.